behaviour

learning strategies, social learning, network-based diffusion analysis, otters, long-term memory

**Author for correspondence:**
Alexander M. Saliveros
e-mail: ams243@exeter.ac.uk

# Learning strategies and long-term memory in Asian short-clawed otters (*Aonyx cinereus*)

Alexander M. Saliveros[1], Eleanor C. Blyth[1],
Carrie Easter[2], Georgina V. Hume[1], Fraser McAusland[1],
William Hoppitt[3] and Neeltje J. Boogert[1]

[1]Centre for Ecology and Conservation, College of Life and Environmental Sciences, University of Exeter, Penryn, Cornwall TR10 9FE, UK
[2]School of Biology, University of Leeds, Leeds LS2 9JT, UK
[3]Department of Biological Sciences, Royal Holloway University of London, Egham TW20 0EX, UK

AMS, 0000-0002-8680-790X; CE, 0000-0001-8538-2076

Social learning, where information is acquired from others, is taxonomically widespread. There is growing evidence that animals selectively employ 'social learning strategies', which determine e.g. when to copy others instead of learning asocially and whom to copy. Furthermore, once animals have acquired new information, e.g. regarding profitable resources, it is beneficial for them to commit it to long-term memory (LTM), especially if it allows access to profitable resources in the future. Research into social learning strategies and LTM has covered a wide range of taxa. However, otters (subfamily Lutrinae), popular in zoos due to their social nature and playfulness, remained neglected until a recent study provided evidence of social learning in captive smooth-coated otters (*Lutrogale perspicillata*), but not in Asian short-clawed otters (*Aonyx cinereus*). We investigated Asian short-clawed otters' learning strategies and LTM performance in a foraging context. We presented novel extractive foraging tasks twice to captive family groups and used network-based diffusion analysis to provide evidence of a capacity for social learning and LTM in this species. A major cause of wild Asian short-clawed otter declines is prey scarcity. Furthering our understanding of how they learn about and remember novel food sources could inform key conservation strategies.

# 1. Introduction

Learning is a process through which experiences change an organism's behaviour so that it is better adjusted to its environment [1]. The degree to which animals rely on social cues when learning has been shown to vary [2–4]. At one end of the spectrum individuals may learn through personal experiences via processes such as trial and error [3,5], and at the other, animals learn from information acquired from others or their products [2–4,6]. Learning socially from the behaviour of others has been described in various taxa, including fish [7–9], reptiles [10,11] and birds [12–14], with the most widely known examples coming from primates [15,16]. On face value, social learning would appear to be universally adaptive, as it negates the potential costs of learning from privately gathered information, such as the increased energy expenditure and/or predation risk associated with trial-and-error learning [4]. However, indiscriminate social learning is unlikely to be adaptive, as information gathered from others can be unreliable or outdated [3,17].

Instead of blindly copying others, animals use 'social learning strategies', which determine, for example, *when* they should copy others instead of learning asocially ('when' strategies), and *who* they should copy ('who' strategies; [3,4,18]). 'When' strategies arise due to the trade-off between the reliability of information and the cost of asocial learning: animals will turn to social learning when costs of asocial learning or using established behaviours are higher than the potential cost of acting on possibly unreliable/outdated information gathered from others [4,19–21]. 'Copying when asocial learning is costly' is perhaps the most documented 'when' social learning strategy to date, and hence potentially the most widespread, with evidence coming from various species of insect [22–24], fish [20,25], bird [26–28] and mammal [29,30].

Most research on when animals employ social learning has been conducted in a foraging context [31]. When animals forage they must decide: (i) where to look for food, (ii) which items are palatable versus unpalatable or even toxic, (iii) how to overcome any protective measures (e.g. a hard exoskeleton or chemical defences), and (iv) when to move on from a particular food patch [31]. During these decision-making processes, learning strategies can greatly affect individuals' efficiency and success. For example, bumblebees (*Bombus impatiens*) rely on scent cues left by conspecifics to identify flowers that have recently had their nectar drained, when the complexity of the flower's morphology means the time and energetic costs of sampling the flower for themselves are higher [24]. Similarly, callitrichid monkeys copied the solving method of their group mates when learning to extract food rewards from complex foraging tasks, where the time and energetic costs of trial and error became prohibitive [29,32].

Where learned information continues to be useful for resource acquisition in the future, it is expected to be committed to long-term memory (LTM; [33–38]). LTM is the storage and recall of information or events over extended periods of time, such as several weeks, months or even years [39]. Upon the discovery and exploitation of novel food sources, animals would benefit from remembering any specific techniques that they learned to overcome any protective measures of the food items. In doing so they would increase their foraging efficiency, and consequently their chances of survival when they return to exploit the same newly discovered resources [36–38]. Research into the LTM of foraging information and behaviours has mainly focused on spatial memory [33–35,40–43], with some other studies investigating memory of food item palatability [44,45]. There have been comparatively few studies examining whether animals remember specific foraging techniques, however (although see [12,36–38]).

Otters (subfamily Lutrinae) face many foraging challenges that would seem to necessitate effective learning mechanisms, strategies and LTM. Sea otters (*Enhydra lutris*), for example, use rocks to break open the shells of clams. Indeed, they are a species renowned for using extractive foraging techniques to overcome the protective measures of their prey [46]. However, practical constraints mean it is difficult to empirically investigate how these otters learn this skill and commit it to their LTM. In fact, the learning strategies and LTM capabilities of otters had gone relatively understudied until recently, despite the wide diversity of foraging behaviours and social group organizations observed across the 13 species [47]. Both smooth-coated (*Lutrogale perspicillata*) and giant otters (*Pteronura brasiliensis*), for example, occur in mixed-sex family groups in freshwater river systems and cooperatively hunt fish [47–49], while Eurasian otters (*Lutra lutra*) forage solitarily in marine and freshwater environments for crustaceans and fish [47,50–52]. Such a diverse range of foraging behaviours and social structures makes otters ideal candidates for research into the evolutionary and ecological drivers of learning strategies, as well as the LTM of such strategies, in a foraging context. Furthermore, 7 of the 13 otter species are categorized as 'vulnerable' or 'endangered' under the International Union for Conservation

of Nature (IUCN) Red List of Threatened Species, due primarily to reduced prey availability [53–59]. Increasing our understanding of how otters learn about novel food sources is, therefore, vital for developing effective conservation efforts.

Of all the otter species, Asian short-clawed otters (*Aonyx cinereus*), native to freshwater swamps and shallow slow-moving rivers in southeast Asia [47,53,60,61], are especially suited for investigations into otter learning strategies in a foraging context. Despite living in family groups of up to 15 individuals, Asian short-clawed otters do not appear to forage cooperatively [47,53,60]. Although there is some suggestive evidence that they cooperatively solve novel food puzzles in captivity [62], in the wild, they independently forage for crustaceans and molluscs [47,53,60]. They overcome the hard shells of their natural prey to extract the palatable meat within by crunching through them with their jaws, or by prying them open with their paws [47,53,60]. However, given that otters forage in river systems and wetlands within view of their group mates [47,53,60], it may be that they attract each other to profitable foraging patches, and observe the food extraction techniques of their group mates.

The first study of social learning in otters reported that smooth-coated otters readily use social learning to solve novel extractive foraging tasks, but no such evidence was found for Asian short-clawed otters [63]. However, this 'negative result' for the latter species might have been due to the small sample sizes of each group (family groups contained five or six individuals), the absence of pups (aged less than 1 year) and sub-adults (aged less than 2 years; [64]), and insufficient data resolution for the task-solving events. Therefore, in this study, we refined the methodology adopted by Ladds *et al.* [63] and presented five novel foraging tasks of varying difficulty to larger captive family groups of Asian short-clawed otters that contained individuals ranging in age from pups to sub-adults to adults. We hypothesized that: (i) Asian short-clawed otters would employ social learning when first interacting with and solving novel foraging tasks, (ii) social learning would be more readily employed when tackling more complex novel foraging tasks where asocial methods are unproductive, and finally (iii) otters would remember how to solve tasks after not having been exposed to them for several months.

# 2. Methods

## 2.1. Study population

All data were collected between November 2017 and May 2018 from three Asian short-clawed otter groups housed at Newquay Zoo (Cornwall, UK—50.41° N, 05.07° W), Tamar Otter and Wildlife Centre (Cornwall, UK—50.69° N, 04.42° W) and New Forest Wildlife Park (Hampshire, UK—50.89° N, 01.50° W), respectively. Henceforth, the wildlife centres will be referred to as 'Newquay', 'Tamar' and 'New Forest'. The groups at Newquay and Tamar were family groups consisting of two parents and their offspring ($n = 12$ individuals each; subsequently $n = 8$ individuals at Newquay, see §2.3 below), whereas the group at New Forest consisted of siblings only ($n = 5$ individuals; electronic supplementary material, table S1).

## 2.2. Experimental procedure

We first collected social association data to create a social network for each of the otter groups, and then presented each group with a series of five novel foraging tasks containing a food reward. After a period of three to five months (depending on wildlife centre constraints), we presented otters with the same tasks for a second time. There were minor changes in the composition and group size of the Newquay and New Forest groups between the two rounds of task presentations. For these groups social association data were collected twice, to form association networks for each iteration (figure 1).

## 2.3. Association networks

We first collected social association data to generate social networks, by observing each group for 15 separate 1 h periods over 10 consecutive days, during which we recorded which otters were associating every 5 min (following [63]). Otters within one body length of one another were considered to be associating, and we identified individuals using distinct facial features and differences in body size [63,65]. We then created a matrix of the total number of 5 min samples when each potential dyad in each otter group was associating, and calculated an association index for each of these dyads using the simple ratio index [63,66]. The total number of 5 min samples where two

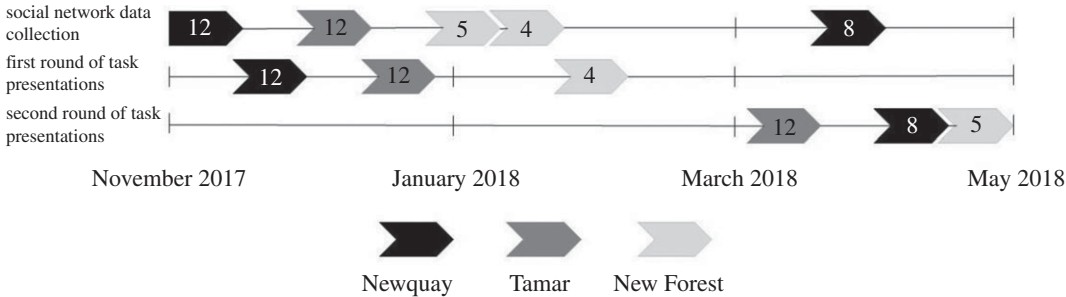

**Figure 1.** A timeline of the data collection period, from November 2017 to May 2018. The top row indicates the timing of social network data collection for each of the Asian short-clawed otter groups (black = Newquay, dark grey = Tamar, light grey = New Forest), the middle row shows the timing of round 1 of the task presentations, and the bottom row specifies the timing of round 2 of the task presentations. The numbers in each of the arrows indicate the number of otters in each group at the time the data were collected.

otters were associating was divided by the sum of (i) the total number of 5 min samples where those individuals were *not* associating plus (ii) the total number of 5 min samples where the two otters *were* associating [66]. The association indices for each dyad within each group formed the association network for that group. While the 5 min sampling points in each of the 1 h sampling periods were not independent of each other, the number of sampling points used to generate association indices, and consequently the network, is not included in the network-based diffusion analysis that we used to test for social learning (see §2.6 below; [63,67]). Thus, there is no risk of sampling point non-independence somehow biasing the results [63,67].

Initially, the group at New Forest consisted of five individuals, and 15 h of association data were collected. However, prior to the first presentation of the novel extractive foraging tasks to this group, one individual was removed due to illness. We, therefore, collected association data for another 8 h to assess whether this removal affected the associations between the remaining four group members (another 15 h of association data could not be collected due to time constraints). A Mantel test (R package 'ade4'; [68]) showed that the association matrix generated for the five-otter group did not significantly correlate with the association matrix for the four-otter group (Mantel test; $r_m = 0.14$, $p = 0.54$, 9999 permutations). We, therefore, used the 8 h, four-otter social network to analyse the data for the first round of task presentations to the four-otter group. For the second round of task presentations, the ill individual had fully recovered (after a one-week absence) and had returned to the group, so we used the 15 h, five-otter network to analyse the data (see §1.1 in electronic supplementary materials for more information).

The Newquay group composition also changed from 12 individuals at the time of the first round of task presentations to eight individuals prior to the second round (one sub-adult and one pup died, and two sub-adults were transferred to a different zoo). We, therefore, collected 15 h of social association data prior to each round of task presentations for this group and used the 12- and 8-otter social networks for analysis of the first and second round of task presentations, respectively (electronic supplementary material, table S1; figure 1). To check that the otter social networks were structured and individuals varied in the degree to which they associated with each other, we measured the heterogeneity of the associations between otters for each network. We calculated each network's social differentiation coefficient by dividing the standard deviation of all the association indices by the mean of all the association indices. The resulting social differentiation coefficients indicated that the networks had a heterogeneous structure (Newquay 12-otter network: 0.48; Newquay eight-otter network: 0.41; Tamar: 0.36; New Forest four-otter network: 0.52; New Forest five-otter network: 0.58). In other words, all group members associated with each other, but the degree of association between individuals varied (electronic supplementary material, figures S1 and S2).

## 2.4. Do Asian short-clawed otters copy each other when first interacting with, and solving, novel foraging tasks?

### 2.4.1. First round of task presentations

We presented each otter group with a series of five novel extractive foraging task types, in the week following the collection of the association data (figure 2; electronic supplementary material, table S2).

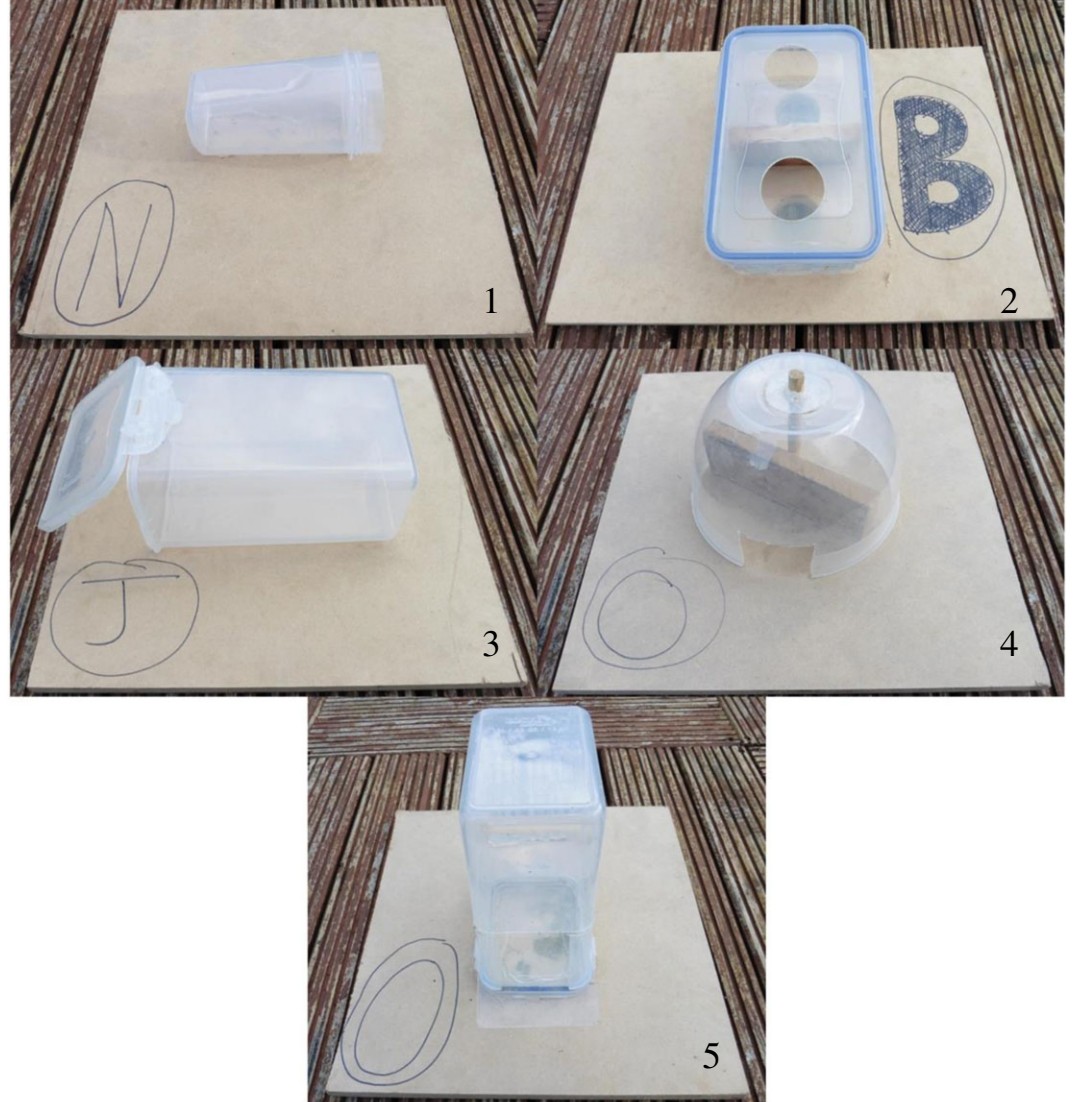

**Figure 2.** The novel foraging task types presented to each group of Asian short-clawed otters. Tasks were baited with a desirable food reward (one 15 g raw beef mince meatball per apparatus) and were numbered based on the assumed difficulty otters would have accessing this reward, with Task 1 assumed to be the easiest and 5 assumed to be the hardest to solve. Task dimensions and details of how food rewards could be accessed from each task type are in electronic supplementary material, table S2.

Each task apparatus was baited with one 15 g raw beef mince meatball. To ensure otters were motivated to solve the tasks and to negate any risk of overfeeding, the equivalent weight of the food used for the rewards in each task was removed from the otters' morning feed. We designed the tasks (based on apparatus from [69]) to resemble the type of foraging interactions that Asian short-clawed otters might experience in the wild when extracting meat from crabs and molluscs, which they access by reaching into rock crevasses and digging through silt [60]. The five task types varied in assumed difficulty and were numbered accordingly, with '1' being assumed the easiest and '5' being the hardest to solve (figure 2). To prevent any possible order effects, we presented tasks types in a random order to each group. We presented a different task type each day for five consecutive days, generating five 'diffusions' of the solutions to the five task types per group (following [63]). On each test day, for each task type, we placed an abundance of identical task apparatus (labelled A–O; figure 2) into the enclosure (always three more apparatus than the number of otters in the group) so that all the otters had the chance to solve a task and to observe group members solving the tasks [63].

We presented the task apparatus at approximately 10.00 each day, between the otters' regular morning and midday feeds. Before each task presentation, all otters were locked in a separate area of the enclosure by the wildlife centre's animal keepers, so that we could lay out the task apparatus in

the main enclosure. The start of each task presentation trial was marked by the otters being released from the locked area into the main enclosure. The task apparatus were left in the enclosure until they had all been solved (i.e. all the food rewards had been removed). We filmed each task presentation trial from two different angles with two Panasonic HC-V380 camcorders to ensure every task apparatus and otter task interaction was captured on film. For each task presentation trial, we reviewed the footage to record the latency (to the nearest second) from the start of the task presentation at which otters initially interacted with, and subsequently solved task apparatus, as well as the identities of the individuals interacting with or solving tasks [63]. Initial interactions were defined as the first physical contact an individual had with a task apparatus, while an otter was considered to have solved a task when it successfully extracted the food reward from a task apparatus [63]. We then mapped these task interaction- and solving-data onto the social association networks (see §2.3 above) using network-based diffusion analysis (see §2.6.1 below) to test whether otters learned from each other to interact with and solve the tasks [6].

## 2.5. Do Asian short-clawed otters remember how to solve novel foraging tasks?

### 2.5.1. Second round of task presentations

Next, we investigated whether the otters would remember how to solve the tasks after several months of not having been exposed to them, or whether they would rely again on social learning to solve them. We presented the exact same task types again in exactly same way as described above, but following a break of 153 days at Newquay, 116 days at New Forest, and 111 days at Tamar (constrained by access to each wildlife centre's otter populations).

## 2.6 Data analysis

All statistical analyses were conducted in R v. i368 4.0.2 [70].

### 2.6.1. Do Asian short-clawed otters copy each other when interacting with, and solving, novel foraging tasks?

The data from all three groups were modelled together using the time of acquisition diffusion analysis (TADA) variant of network-based diffusion analyses (NBDA; R package 'NBDA'; [6,71]) to investigate whether the otters learned from each other to interact with and solve the novel foraging tasks in each task presentation round. TADA infers the spread of information through social transmission when the time sequence in which individuals first exhibit novel behaviours, in this case interacting with or solving the tasks, follows associations between individuals in a group's association network [6,63,72]. We used a model comparison approach, comparing social transmission models with asocial learning models, while also controlling for individual-level variables (ILVs) that might have affected the rate of learning. Fitting parameters that control for the effects of ILVs on asocial and social learning rates, allows NBDA to assess how these learning strategies may interact [6,73]. The ILVs we included were each otter's age (in years) and sex (females coded as '1' and males coded as '0'). The behaviour and body size of Asian short-clawed otter pups changes dramatically over the first year of their life [64,74]. Here, pups varied in age from three to nine months old between groups and task presentation rounds, and were, therefore, at different stages in their development (electronic supplementary material, table S1; [64,74]). In order to account for this, the age of otters less than 1 year old was included as a proportion of 1 year (e.g. the age of a three-month-old pup was denoted as 0.25 in the analysis).

Models fitted with a parameter that assumes ILVs only affect asocial learning rates are known as additive models (because the asocial and social learning rate estimates are summed; [6,73]). Therefore, where additive models best fit the data, social and asocial learning are inferred to be independent stochastic processes, and individuals acquire new behaviours through either social or asocial learning. By contrast, multiplicative models are fitted with parameters that assume ILVs affect both asocial and social learning rates (and the asocial and social learning rate estimates are multiplied), so when these models best fit the data, new behaviours are inferred to be acquired via a combination of both learning methods [6,73]. If models without ILVs are best-supported by the data, this would indicate that neither asocial nor social learning rates are affected by ILVs. We fitted additive, multiplicative and no-ILV versions of each model with a parameter that represented social transmission rates to be either the same or different across task types. If social transmission rates increased with task difficulty,

this would indicate otters relied more on social learning to interact with/solve more difficult tasks and so would provide support for the 'copy when asocial learning is costly' strategy [4].

All types of social learning models (i.e. additive, multiplicative and no-ILV models where social transmission was constrained to be either the same or different between task types) fit with various combinations of ILVs, were fit four times for each behaviour (i.e. interacting with, or solving tasks) in each presentation round: twice with a constant baseline asocial learning rate, and twice with a baseline function corresponding to a gamma distribution to the baseline rate of asocial learning [6,73]. This latter baseline function was used to control for possible changes in asocial learning rates over time during each diffusion [6,73]. One set of constant baseline models and one set of gamma baseline models were fit to the association networks described above, here on referred to as 'social networks', where associations between individuals were heterogeneous [6,73,75,76]. Another set of constant and gamma baseline models were fit to networks where all association indices between individuals were set to be equal to 1, and therefore homogeneous; these are referred to as 'group networks' [75–78]. This was to determine whether, if social learning was present, the novel behaviours diffused via the strength of associations in each group's social network or evenly through the group [75–78].

Akaike weights for each individual model were summed to quantify the relative support for each type of model, with each baseline rate of asocial learning, fit to each network. Summed Akaike weights were then used to compare asocial versus social learning models, and social learning models with the same versus different rates of social transmission across the task types. We used Akaike's information criterion (AIC) to select the model with the lowest AIC value that best explained the data of all the models considered (see [73] for detailed model selection procedure; [79]). The estimates from this model of the rates of social transmission per unit of network connection relative to the baseline rate of asocial learning (from here onwards referred to as 'social transmission rate') are reported (see §1.2 in electronic supplementary materials).

### 2.6.2. Do Asian short-clawed otters remember how to solve novel foraging tasks?

To test whether otters remembered how to solve the tasks after not having seen them for several months, we tested whether they took less time to solve them in the second round of task presentations compared with the first round. We used a generalized linear mixed-effects model (GLMM) with a gamma error structure and log link function, as the data were positively skewed (R package 'lme4'; [80]). The model fitted latency (in seconds) from the first time each otter interacted with an apparatus to the time they first solved that task type as the response variable. Fixed effects included: the presentation round ('1' or '2'), otter age (in years—age of the pups was expressed as a proportion of year; see above), sex (females coded as '1' and males coded as '0') and task type (1–5, based on assumed difficulty; figure 2). Individual otter identity was included as the random effect to control for repeated measures of the same individuals. All combinations of fixed effects were assessed and ranked by AIC via subset selection of the maximal model (the model containing all fixed effects; R package 'MuMIn'; [81]). Models within ΔAIC ≤ 2 of the best-supported model (i.e. that with the lowest AIC value) formed the 'top set', after more complex versions of nested models had been removed in order to retain simpler models with stronger weights ('nesting rule'; [82]). Results from the best-supported model are reported as the top set was formed of more than one model [79].

## 3. Results

## 3.1. Do Asian short-clawed otters copy each other when first interacting with, and solving, novel foraging tasks?

### 3.1.1. Task interaction—round 1

When the tasks were first presented to the otter groups, all the otters, barring two of the pups (less than 1 year) in the Newquay group, interacted with at least three of the five task types (electronic supplementary material, table S3). The best-supported models were multiplicative models with gamma baselines fit to the social networks, where the rate of social transmission was the same between task types (Akaike weight: 67.13%; table 1). This indicates that, in the first round of task presentations, otters tended to start interacting with a specific task sooner once an individual closely associated with them started doing so, suggesting that at least some otters were influenced by social

**Table 1.** A comparison of the relative support (i.e. percentage of overall support based on summed Akaike weights) for different social and asocial learning models with gamma baselines, fit to group (homogeneous associations between individuals) and social (heterogenous associations between individuals) networks, for the first instance each otter interacted with, and solved, novel foraging tasks during each round of task presentations (see electronic supplementary material, table S9 for the full table including the relative supports of models fit with constant baselines). *Italicized* values indicate the model types with the most statistical support for each behaviour in each presentation round.

| model type | rate of social transmission between tasks | round 1 | | round 2 | |
| --- | --- | --- | --- | --- | --- |
| | | interaction | solve | interaction | solve |
| asocial | | <0.01 | 0.24 | <0.01 | 10.09 |
| group network | | | | | |
| additive | same | 0.07 | 1.21 | <0.01 | 3.15 |
| | different | 0.05 | 0.06 | <0.01 | 2.02 |
| multiplicative | same | 23.94 | 0.85 | 0.01 | 3.31 |
| | different | 2.47 | 10.76 | 7.26 | 28.83 |
| no ILVs | same | <0.01 | <0.01 | <0.01 | <0.01 |
| | different | <0.01 | <0.01 | <0.01 | <0.01 |
| social network | | | | | |
| additive | same | 1.25 | 1.15 | <0.01 | 3.15 |
| | different | 0.24 | 2.54 | 0.74 | 2.26 |
| multiplicative | same | *67.13* | 4.51 | 0.07 | 3.40 |
| | different | 4.84 | *79.77* | *88.68* | *41.09* |
| no ILVs | same | <0.01 | <0.01 | <0.01 | <0.01 |
| | different | <0.01 | <0.01 | 0.01 | <0.01 |

learning. The best-supported model (electronic supplementary material, table S4), estimated the social transmission rate to be 5.51 (95% CI = [2.14, 13.96]; table 2; figure 3*a*) across all task types. This equates to the otters (except for the first otter to interact with a task in each diffusion, i.e. the 'innovator') using social information when carrying out 88.07% (95% CI = [75.61, 94.71]; table 3) of initial task interactions. Additionally, the best model denoted that older otters tended to start interacting with each task sooner, with the rate of first interaction increasing by 1.17× (95% CI = [1.12, 1.22]) per year of age.

### 3.1.2. Task solving—round 1

Most individuals solved at least one task type, except for the pups (less than 1 year; $n = 7$), who did not solve any tasks (electronic supplementary material, table S3). The best-supported models were multiplicative models, with gamma baselines fit to the social networks, where the rate of social transmission was different between task types (Akaike weight: 79.77%; table 1). This suggests that, like with their first task interactions (above), otters' first task solves were attributable to a mix of both social and asocial learning. However, it is worth noting that the range of the 95% confidence intervals for the percentage of task solves which occurred due to social learning ranged from 26.7% (Task 1) to 82.74% (Task 4; table 3). So although the best model suggests otters do use social learning to solve novel tasks in the first round of presentations, the extent to which they do so is not entirely clear. The social transmission rate estimates from the best-supported model (electronic supplementary material, table S5) varied across task types, but they did not systematically increase or decrease with task difficulty (table 2; figure 3*b*). Accordingly, the percentage of first task solves in which otters used social information also varied among tasks but did not increase or decrease with task difficulty (table 3). Lastly, the best model also indicated that older individuals solved each task sooner (1.15×; 95% CI = [1.09, 1.18]; per year of age), and that females tended to be slower to solve the tasks than males (0.62×; 95% CI = [0.36, 1.03]).

**Table 2.** Social transmission rates estimated by the best-supported model when otters were interacting with, and solving, tasks in each task presentation round.

| presentation round | interaction/solve | estimated social transmission rate for each task type (95% CI) | | | | |
|---|---|---|---|---|---|---|
| | | 1 | 2 | 3 | 4 | 5 |
| 1 | interaction | 5.51 (2.14, 13.96)[a] | 5.51 (2.14, 13.96)[a] | 5.51 (2.14, 13.96)[a] | 5.51 (2.14, 13.96)[a] | 5.51 (2.14, 13.96)[a] |
| 1 | solve | 4.71 (1.61, 12.18) | 1.76 (0.17, 5.73) | 0.29 (0.00, 2.31) | 1.43 (0.08, 5.58) | 0.27 (0.00, 2.35) |
| 2 | interaction | 1.09 (0.00, 5.13) | 2.61 (0.55, 7.75) | 3.26 (0.80, 9.41) | 6.06 (1.99, 15.52) | 8.33 (2.61, 31.00) |
| 2 | solve | 1.52 (0.17, 4.68) | 2.02 (0.60, 5.21) | 0.01 (0.00, 1.33) | 0.00 (0.00, 0.58) | 0.50 (0.00, 2.58) |

[a]The social transmission rate parameter estimate for initial task interactions in the first task presentation round is constrained to be the same across all tasks, as denoted by the best-supported model.

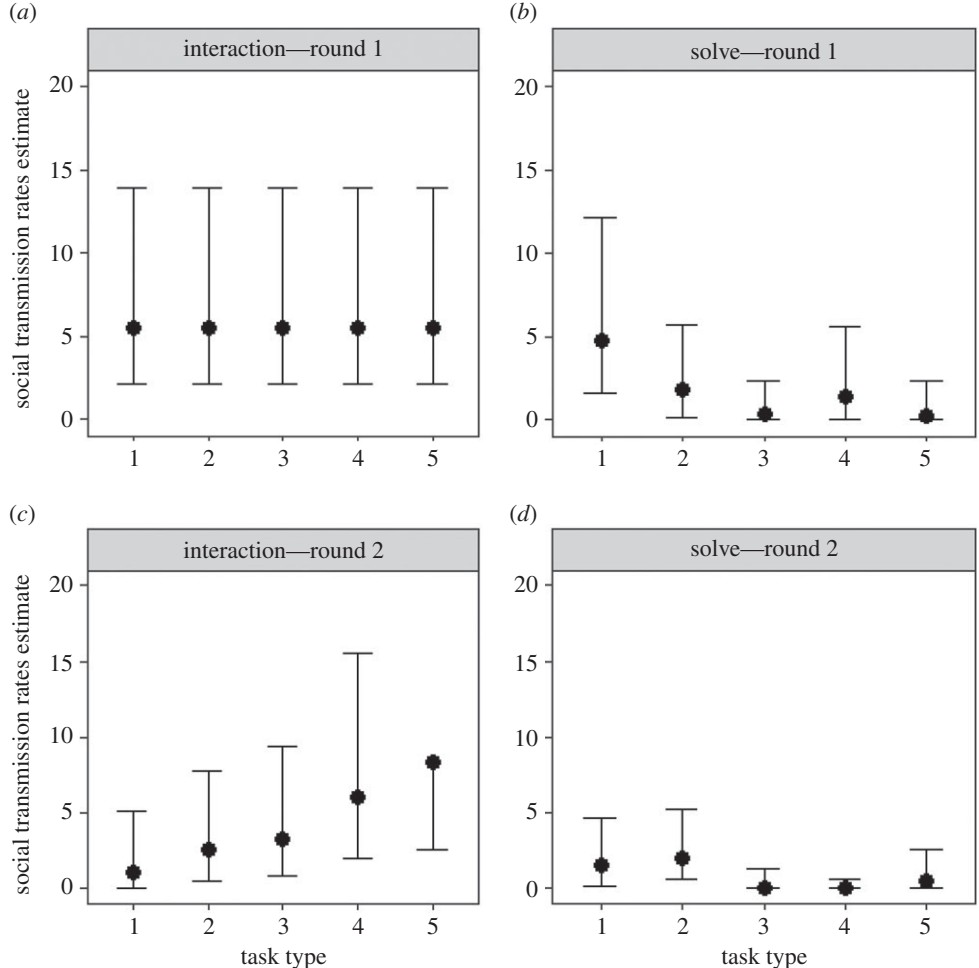

**Figure 3.** The estimated social transmission rates per unit network connection relative to the baseline rate of asocial learning for task types 1–5, as estimated by the best-supported models for: (*a*) task interaction data in the first round of task presentations (where the social transmission rate parameter estimate is constrained to be the same across all tasks); (*b*) task solving data in the first round of task presentations; (*c*) task interaction data in the second round of task presentations; (*d*) task solving data in the second round of task presentations, when all three groups were modelled together. Error bars indicate 95% confidence intervals. The upper 95% confidence interval of the social transmission rate estimate for when otters were interacting with task type 5 in the second task presentation round is 31.00 (table 2).

## 3.2. Do Asian short-clawed otters copy each other when interacting with, and solving, foraging tasks they have encountered before?

### 3.2.1. Task interaction—round 2

Similarly to when the five task types were presented to the otter groups initially, when presented for a second time, all otters interacted with at least three task types (electronic supplementary material, table S6). The best-supported models were multiplicative models, with gamma baselines fit to the social networks, where the social transmission rate was different between task types. This suggests that, where otters relied on social learning, they tended to start interacting with a specific task once an individual closely associated with them started doing so (Akaike weight: 88.86%; table 1). Social transmission rates estimated by the best model (electronic supplementary material, table S7) appeared to increase with task difficulty (table 2; figure 3*c*). The percentage of first interactions that were due to social learning also increased with task difficulty (table 3). Additionally, the best model denoted that older otters tended to start interacting with each task sooner, with the rate of first interaction increasing by 1.12× (95% CI = [1.06, 1.18]) per year of age.

### 3.2.2. Task solving—round 2

In the second round of task presentations, most otters solved at least two task types (electronic supplementary material, table S6). The best-supported models were multiplicative models with

**Table 3.** Estimated percentage of first task interactions and solves that occurred due to social learning (excluding the innovator) in each task presentation round.

| presentation round | interaction/solve | estimated percentage of acquisition events that occurred due to social learning for each task type (95% CI) | | | | |
|---|---|---|---|---|---|---|
| | | 1 | 2 | 3 | 4 | 5 |
| 1 | interaction | 88.07 (75.61, 94.71) | 88.07 (75.61, 94.71) | 88.07 (75.61, 94.71) | 88.07 (75.61, 94.71) | 88.07 (75.61, 94.71) |
| 1 | solve | 82.55 (65.53, 92.23) | 61.25 (7.12, 68.21) | 23.89 (0.00, 71.28) | 59.54 (4.40, 87.14) | 22.63 (0.00, 70.27) |
| 2 | interaction | 54.85 (0.00, 84.22) | 78.61 (39.50, 82.03) | 83.31 (59.80, 95.09) | 90.16 (78.17, 97.81) | 92.50 (82.27, 99.88) |
| 2 | solve | 60.00 (15.35, 81.58) | 69.40 (38.35, 78.80) | 1.01 (0.00, 55.74) | 0.00 (0.00, 33.06) | 37.13 (0.00, 75.14) |

**Table 4.** Model selection table for variables affecting latency (in seconds) from the first time each otter interacted with an apparatus to the time they first solved that task type. Variables included as fixed effects in the models were the task presentation round (i.e. '1' or '2'), otter age (in years), sex (females coded as '1' and males coded as '0'), and task type (1–5, based on assumed difficulty). Retained top set models and adjusted weights ranked by AIC value after selection for $\Delta AIC \leq 2$ and application of the nesting rule. As there were two models in the top set, the findings from the top-ranked model (*italicized*) are reported in the text.

| fixed effects | intercept | d.f. | logLik | AIC | ΔAIC | adj. weight |
|---|---|---|---|---|---|---|
| *presentation round + sex + task type* | *2.20* | *9* | *−660.39* | *1338.76* | *0.00* | *0.65* |
| presentation round + task type | 2.43 | 8 | −662.00 | 1339.98 | 1.22 | 0.35 |
| 1 | 4.87 | 3 | −722.18 | 1450.53 | 111.77 | 0.00 |

gamma baselines fit to the social networks, where the rate of social transmission was different between task types (Akaike weight: 41.09%; table 1). This suggests that the first task solves achieved by the otters in the second round of task presentations were attributable to a mix of both social and asocial learning. However, it is difficult to say for sure whether otters copied their close associates. Multiplicative models with gamma baselines, where social transmission rates were different between tasks that were fit to the social networks, only had 1.43× more support than those fit to the group networks where all individuals are assumed to have equal association strengths (Akaike weight: 28.83%; table 1). Furthermore, the best model that was fit to these group networks had a ΔAIC within 2 of the overall best model which was fit to the social networks (electronic supplementary material, table S8). The social transmission rate estimates from the best-supported model (electronic supplementary material, table S8) varied across task types, but they did not consistently increase or decrease with task difficulty (table 2; figure 3*d*). The percentage of first task solves in which otters used social information also varied between tasks but did not increase or decrease with task difficulty either (table 3). Indeed, the range of the 95% confidence intervals for the percentage of task solves which occurred due to social learning ranged from 33.06% (Task 4) to 75.14% (Task 5; table 3). So although the best model suggests otters used social learning to solve tasks, the extent to which they did so is unclear. Finally, the best model indicated that older individuals solved each task sooner, with the rate of solving increasing by 1.14× (95% CI = [1.08, 1.21]) per year of age.

## 3.3. Do Asian short-clawed otters remember how to solve novel foraging tasks?

The latency between the first time otters interacted with an apparatus to the time they first solved that task type was affected by task presentation round, task type and the otters' sex (table 4). On average, otters solved tasks faster in the second round of presentations (mean ± standard deviation of the fitted solve latencies from the top-ranked model; 65.23 ± 70.06 s) than they did in the first task presentation round (160.64 ± 156.83 s; figure 4*a*). It took otters longer on average to solve tasks that were assumed to be more complex across both presentation rounds (mean ± standard deviation of the fitted solve latencies from the top-ranked model; Task 1: 10.14 ± 7.83 s; Task 2: 28.12 ± 21.96 s; Task 3: 198.30 ± 151.50 s; Task 4: 247.83 ± 135.61 s; figure 4*b*). Task 5, which was assumed to be the most difficult to solve, was the exception to this, however, as it took otters the third longest to solve on average (135.53 ± 76.02 s; figure 4*b*). Across both task presentation rounds, males solved tasks slightly faster (mean ± standard deviation of the fitted solve latencies from the top-ranked model; 108.57 ± 117.08) than females (128.03 ± 154.08 s; figure 4*c*).

## 4. Discussion

Here, we present the first empirical evidence we are aware of that Asian short-clawed otters exhibit social learning when first encountering novel foraging challenges. In addition, our results suggest that they may also remember how to overcome these foraging challenges months later.

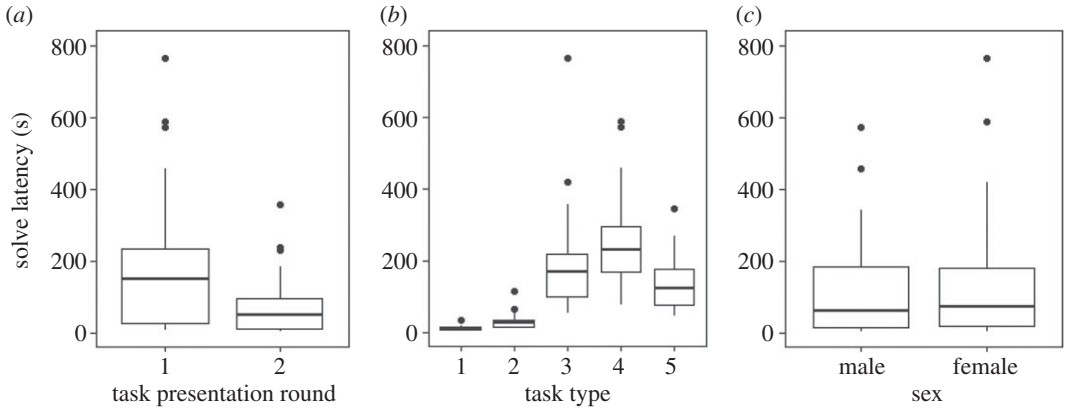

**Figure 4.** The fitted values from the best-supported GLMM showing how the latencies (in seconds) between the first time otters interacted with task apparatuses to the time they first solved that task type were affected by (*a*) task presentation round, (*b*) task type and (*c*) otters' sex. The bold line within each box indicates the 50th percentile, the top and bottom of each box signify the 75th and 25th percentiles, respectively. The whiskers signify the highest and lowest values that are not outliers, while outliers are represented by the points above the whiskers.

## 4.1. Do Asian short-clawed otters copy each other when first interacting with, and solving, novel foraging tasks?

When encountering novel foraging tasks, otters were found to use a combination of social and asocial information when deciding to start interacting with them. Their use of social learning here is consistent with 'stimulus enhancement', i.e. when the observation of individuals engaging with an object causes those who observed the behaviour to direct their own behaviour to the same, and/or similar, objects [2,3]. Stimulus enhancement is widespread among socially learning species (including: bumblebees, *Bombus terrestris*, [83]; Florida red-bellied cooters, *Pseudemys nelson*, [11]; greylag geese, *Anser anser*, [84]; domestic goats, *Capra hircus*, [85]; spotted hyenas, *Crocuta crocuta*, [86]; and long-tailed macaques, *Macaca fascicularis*, [87]) and is, therefore, thought to be adaptive [88]. For example, meerkats that observed conspecifics interacting with and solving extractive foraging tasks, interacted with those tasks more persistently at higher rates, which resulted in increased solving success [89]. Thus, stimulus enhancement increased the foraging success of the observers. Similarly, the otters we studied appeared to start interacting with novel tasks when they had observed their close associates interacting with them, suggesting that their likelihood of exploiting these novel resources was facilitated through stimulus enhancement.

Otters were also found to use a combination of social and asocial learning when solving tasks for the first time. However, social transmission parameter estimates were generally lower than when otters were first interacting with tasks. This could suggest that social information is relied upon more when otters are learning to locate and/or interact with novel tasks, but that once initial contact has been made, they are less reliant on social information when learning to actually solve them. Consequently, we posit that in the wild these otters learn how to exploit food resources mainly asocially, but they may learn where to find food, or what type of food type to exploit, by observing their conspecifics foraging nearby [47].

## 4.2. Do Asian short-clawed otters remember how to solve novel foraging tasks?

When the tasks were presented to the otters again three to five months after the first presentation, they solved them faster than they did in the first round. Furthermore, similarly to the first task presentation round, estimated social transmission rates were generally lower when otters were solving tasks than when they were interacting with them. Thus, otters again appear to rely less on social learning and more on asocial methods to solve tasks. Studies on the LTM capabilities of other mammal species, such as goats (*C. hircus*; [90]) and common marmosets (*Callithrix jacchus*; [37,38]), have demonstrated that subjects asocially solved food puzzles significantly faster when presented with tasks months or even years after first learning how to do so. Therefore here, where the otters relied on asocial methods to solve tasks more quickly, it suggests that they individually remembered the specific method

required to solve each task months after first learning how to do so. This in turn suggests that Asian short-clawed otters may possess LTM capabilities when it comes to overcoming the protective measures of food items.

In the second round of presentations, the best model indicated that otters used a combination of social and asocial methods when first deciding to interact with tasks. Indeed, estimated social transmission rates were generally higher when otters were interacting with tasks compared with when they were solving them. This suggests that they continued to observe each other to decide when to start interacting with these still somewhat foreign objects. However, once they started interacting with the tasks, they remembered how to solve them asocially. Animals are known to use cues to help them remember information. For example, food-storing bird species often use object and spatial cues to remember the locations of their cache sites [91]. Here, the action of interacting with the tasks may have acted as a cue for the otters and triggered memories about how to solve them, thereby facilitating faster solve times in the second round.

## 4.3. Does task difficulty affect Asian short-clawed otters' reliance on social information?

When otters were solving the tasks, the best models indicated that their reliance on social learning was higher when solving Tasks 1 and 2 in both task presentation rounds, and when solving Task 4 in the first round. The solve latencies across both rounds suggest that Tasks 1 and 2 were the easiest tasks to solve, while Task 4 was the most difficult. These results would suggest Asian short-clawed otters do not only 'copy when asocial learning is costly or unproductive' as they appear to employ social learning when solving the easiest and hardest tasks, compared with when solving the moderately difficult ones (3 and 5). Although, as the estimated social transmission rates for task solves as compared with task interactions are relatively low, it may be that the apparent differences in otters' reliance on social information between tasks when solving, may just have been due to chance and should, therefore, be interpreted with caution.

As discussed above, we found strong evidence for otters using social learning when deciding to first interact with tasks in both presentation rounds. In the first presentation round, otters' use of social information was the same across all tasks, as we would expect given that the otters had not encountered the tasks before, and thus, had no prior knowledge about the difficulty of particular tasks that could influence their reliance on social information. In the second round, however, the otters' use of learning when interacting with tasks increased with assumed task difficulty. This would seem to suggest that otters remembered the relative difficulty of each task and adjusted their reliance on social learning upon recognizing particular tasks. Then, as the estimated social transmission rates when solving tasks in the second round were lower than when otters were deciding to interact with them, it may be that once they interacted with the tasks, they remembered how to solve them on their own (see §4.2 above). However, given the solve latencies indicated Tasks 3 and 4 were harder to solve than Task 5, we would expect otters' social information use when interacting with those tasks to be higher than when interacting with Task 5, which was not the case. Furthermore, we would expect the pattern of social information use when otters were solving tasks in the first round to be reflected in their social information use when deciding to interact with tasks in the second round, which was also not the case. Therefore, our suggestion that otters may remember task difficulty and adjust their reliance on social learning accordingly should be interpreted with caution. Overall, replication of this work with a larger sample size of otter groups, and using tasks with more pronounced differences in difficulty, is needed to definitively determine whether otters turn to their group mates when individual learning becomes too costly.

## 4.4. Do age and sex affect learning and long-term memory in Asian short-clawed otters?

Across both task presentation rounds, older otters appeared to learn faster when they were interacting with, and solving, tasks. Previous work on New Caledonian crows (*Corvus moneduloides*) suggests that animals with prior experience of problem solving quickly learn how to overcome analogous problems [92]; crows that had experienced a task which involved collapsing a platform without the use of stones, quickly learned to drop stones onto a similar platform in order to access a food reward [92]. In our study, all three otter groups had been exposed to foraging tasks as part of the enrichment schemes designed by the wildlife parks, although these tasks were of a simpler design than the ones we presented here (Tupperware boxes with either clip or screw off lids; similar to those used by Ladds *et al.* [63]). Having been at the various facilities for longer, older otters would have experienced

food puzzles more often than younger otters. Indeed, otters less than or equal to 2 years old had never been presented with food puzzles prior to this study. Therefore, we posit that despite our tasks providing a mechanistically new foraging challenge to all the test subjects, older otters may have been familiar enough with the concept of extracting a reward from a puzzle box that they were able to learn how to solve the tasks faster than younger otters with less experience.

Furthermore, across both task presentation rounds only one pup managed to solve tasks (task types 2 and 5 in the second round). This finding is consistent with results of an aforementioned study on meerkats, that noted juveniles were rarely successful at extracting food rewards from novel foraging tasks; only one of 26 juveniles that interacted with tasks obtained a reward [69]. The meerkats' relatively small size in relation to tasks and lack of dexterity compared with adults were cited as possible reasons for this, and we suspect these factors may also explain why the otter pups struggled to solve tasks here. The oldest individuals on the other hand, were usually the first to interact with and solve tasks. It has been widely reported that older individuals often act as innovators and/or demonstrators for new behaviours in other taxa [93–95]. We suggest that in Asian short-clawed otter groups, older individuals are the innovators of new behaviours, and so also act as demonstrators to other group members in foraging scenarios where they employ social learning.

When otters were solving tasks in the first round of presentations, male otters were found to learn slightly faster than females. Moreover, across both task presentation rounds, males also solved tasks marginally faster than females on average. Although empirical evidence is lacking, it is thought that wild Asian short-clawed otter groups may be formed of a dominant female and her offspring, as similar social group-living otter species, such as smooth-coated and giant otters, are organized in this way [47]. However, during the breeding season adult male otters, resident within a group's territory, are temporarily tolerated as part of the group [47]. Given that adult male otters may be mainly solitary in the wild, the small sex differences we observed in learning rate and task solving speed may stem from the fact that they have to be more self-sufficient, as they often do not have group mates to rely on [47].

## 4.5. Wider implications

This study is not without limitations; our experiments focused on captive otter groups, so interpreting and extrapolating the implications of these findings to wild populations should be done with caution. Unfortunately, given their elusive nature, studies on large populations of wild otters in their natural habitat may be difficult. Instead, steps must be taken to ensure findings from work on captive subjects is as applicable to their wild counterparts as possible. In captivity, otters are often given food such as mince meatballs, day-old chicks and sliced fish [96,97], which require no extractive skills to consume. So perhaps future research investigating which learning strategies otters use when first experiencing novel foraging challenges, should present captive otters with natural prey items, such as crabs and molluscs, which do require the use of extractive techniques [60]. Indeed, this may provide sufficient novelty for the captive subjects, yet enough similarity to the challenges faced by wild individuals, to safely interpret and extrapolate any findings to wild populations. Additionally, when generating association networks for each otter group, we recorded interactions throughout their daily routine. Future studies may benefit from observing otters specifically within the context in which they are investigating learning strategies. In past research on starlings, whether social networks were formed from associations observed during perching or foraging behaviour affected whether the spread of novel behaviour by social learning was detected [98]. Therefore, here it may have been beneficial to carry out observations during foraging periods too, as interactions at these times may differ to otters' 'day-to-day' interactions and thus may affect the extent to which social or asocial learning is detected by NBDA.

While this study has delved into the learning strategies of Asian short-clawed otters and presents evidence of social learning and LTM in this species, it only does so in the context of exploiting novel food sources. It may be valuable to expand the knowledge base further and investigate learning strategies and memory of this species in other contexts, such as when learning about predators, and coordinating anti-predator behaviour (e.g. [99,100]). Another group-living otter species, the giant otter, has been observed responding to jaguars (*Panthera onca*) with group-coordinated defence behaviour, namely mobbing [101], a tactic which Asian short-clawed otters are also thought to adopt in the wild [47]. Whether Asian short-clawed otters learn socially about predators and coordinated anti-predator behaviour is the focus of our current research efforts.

Studies of this nature are important for the development of appropriate conservation strategies for species at risk of extinction. Moreover, demonstrating the cognitive abilities and the ways in which otter species or other animal families learn, can promote their significance to the public, who can help to fund conservation efforts to protect them [102]. More specifically, furthering understanding of animal learning strategies may be particularly useful for increasing the success of translocation and reintroduction programmes. In those species that socially learn, it may be beneficial to add a previously introduced individual back into the captive population, from whom they can learn adaptive behaviour before being reintroduced themselves [103]. Research has shown that social learning was enhanced in reintroduced primates, when a skilled demonstrator was included in the group [102]. Additionally, common marmosets directly imitate the task solving techniques of 'knowledgeable' individuals [104]. As our study suggests that Asian short-clawed otters use social learning when exploiting novel food sources, it is plausible that previously reintroduced individuals could act as demonstrators of adaptive foraging techniques to naive otters. In this way, they might learn to successfully obtain natural prey, which is in ever-decreasing supply, thereby promoting the success of reintroduction to the wild.

Given the reduced availability of natural prey [53], it would also be beneficial for both wild otters, and otters set for reintroduction, to learn what kind of new food resources, different from their natural prey, are safe and beneficial to eat. The use of social information by Asian short-clawed otters could potentially be exploited to teach them which novel food sources to feed on, as well as how and where to forage for such prey. Indeed, studies have shown that the highest reintroduction success comes from teaching subjects in as much detail as possible about the challenges they will face in the wild [105,106]. Research on fishers (*Martes pennant*), a mustelid closely related to otters, has shown that although individuals were presented with natural prey items in captivity, they starved to death when reintroduced to the wild, as although they had developed the skills to kill prey, they did not know how to find it [106].

Teaching adaptive behaviour to captive individuals or modifying maladaptive behaviours of wild animals would be fruitless, however, if they are not capable of remembering these new skills. Therefore, research into the memory capabilities of species learning novel behaviours is equally as important as identifying the learning strategies themselves. Evidence presented here suggests that extractive foraging skills learned by Asian short-clawed otters in captivity are remembered for at least several months. However this study, like the majority of other research into the LTM of new foraging techniques, has focused on captive animals [37,90,107], so it is difficult to know to what extent skills learned in captivity would be applied in the wild. Henceforth, future research should aim to investigate whether skills taught in captivity or to wild animals during rehabilitation, are still prevalent months or years after release, in order to determine whether such tactics are successful or worthwhile.

In conclusion, this first evidence of social learning and LTM in Asian short-clawed otters provides important insights into the cognition of this 'vulnerable' species [53]. However, given the diversity of social group organizations and learning strategies, comparative studies of cognitive processes across the different species are crucial to provide insights into how to develop management plans most effectively for the conservation of this, and other otter species.

Ethics. Ethical approval for this work was granted by the College of Life and Environmental Sciences Ethics Committee at the University of Exeter, as well as by Newquay Zoo, Tamar Otter and Wildlife Park and New Forest Wildlife Park. Experiments were designed and executed in accordance with the ASAB/ABS Guidelines for the Use of Animals in Research [108].

Data accessibility. The data and R code used in the NBDA and GLMM analyses are available through the Dryad Digital Repository: https://doi.org/10.5061/dryad.sf7m0cg3s [109]. Model selection tables for the NBDA, and tables detailing otter group compositions, descriptions of the novel foraging tasks, and otter task interactions and solves, as well as figures illustrating otter group association networks, have been uploaded as part of the electronic supplementary materials.

Authors' contributions. The study was designed by A.M.S., E.C.B., G.V.H., F.M. and N.J.B. All observational and experimental data were collected by A.M.S., E.C.B., G.V.H. and F.M. Data analysis was led by A.M.S. and W.H., with support from C.E. and F.M. Manuscript was written by A.M.S. with support from F.M., and was supervised by N.J.B and W.H. who critically revised the manuscript. The manuscript was revised by A.M.S., N.J.B., W.H. and C.E. All authors gave final approval for this publication.

Competing interests. The authors declare no competing interests.

Funding. N.J.B. is funded by a Royal Society Dorothy Hodgkin Research Fellowship.

Acknowledgements. We would like to thank Dr Kathy Baker, Sam Harley, Mike Dowman and the staff at Newquay Zoo, John and Mandy Allen of Tamar Otter and Wildlife Centre, and Jason Palmer and the staff at New Forest Wildlife Park, without whom this research would not have been possible. We would also like to thank the editor and two anonymous reviewers for their constructive feedback that helped to improve the manuscript.

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
