## [Reviewer comments · Royal Society Open Science]

Review History

RSOS-201215.R0 (Original submission)

Review form: Reviewer 1

Is the manuscript scientifically sound in its present form?

Yes

Are the interpretations and conclusions justified by the results?

Yes

Is the language acceptable?

Yes

Do you have any ethical concerns with this paper?

No

Have you any concerns about statistical analyses in this paper?

No

Recommendation?

Accept with minor revision (please list in comments)

Comments to the Author(s)

The authors describe a two-phase experiment studying the social learning of extractive foraging techniques in three groups of captive Asian short-clawed otters. They find good evidence that a tendency to interact with the task spreads socially, but weaker evidence that solving the task spreads in this way, suggesting solving the task relies on asocial learning to be solved. In addition, the otters solved the tasks faster a few months later, suggesting they remembered how to solve them. Older otters were more skilled at solving the tasks, as were males, and the otters probably found the harder tasks harder to solve.

Overall, I think this paper is strong. As the above summary suggests, the authors ask a lot of questions, they nonetheless provide compelling answers through a good experimental design and powerful analytic strategy. I have only minor concerns with the manuscript which are as follows:

lines 28-31 (and throughout). Here the authors set up social and asocial learning as two separate mechanisms. I know this isn't a focus of the paper, but I wonder if the emphasis on the separateness is excessive. Yes, talking about social vs. asocial learning has been wonderfully productive in the scientific literature. But are they this separate at the psychological level, or is this just how we choose to think about it? We could equally ask "how sensitive are animals to social cues?" which is more nuanced (and fits perfectly with the analytical approach). So, no big deal, but maybe tweak the wording a bit?

lines 45-46. This sentence feels like it should be at the start of the next paragraph, not the end of this one.

lines 50-58. This paragraph needs better examples. Those provided are good examples of animals engaging in strategic social learning, but do we have proof this improves their efficiency and success?

lines 93-95. It feels odd that the notion of otters observing each other's behavior isn't raised as a possibility here.

lines 126-128. This is a good method. I'd additionally like a brief note that (1) there is no autocorrelation over the series of observations for each dyad, just so we know the observations are statistically independent (might be tricky to calculate this with only 15 observations though) and (2) there is significant variation between the association strengths of different dyads (i.e. do otters have structured social networks?).

lines 133. Do we really need this equation? It's just describing a proportion. I think it would be easier to stick with words here.

lines 142-143. Are the authors not worried that removing one otter seemed to change the social network so drastically? I'd be worried that the social network is just noise. The authors results suggest social networks are real (they guide information flow), but still, this should probably be discussed in the discussion.

lines 145-147. Given that the social networks are apparently very labile, are the authors not worried that the reintroduction of the recovered otters produced yet another, novel, social network structure?

line 197 - typo: analyses

lines 216-224: This nomenclature (additive vs. multiplicative) feel weird. Why does additive mean that individual effects act only on asocial learning? I can accept that this is the case because of how NBDA works under-the-hood, but it's not clear to me why this is the case.

lines 228-239: This paragraph is really hard to follow. It's clear the authors did a lot of stuff, but I have little idea exactly what, or how this paragraph corresponds to the structure of the results section. What are the types of social learning models? What are the information types? etc.

Figure 3. The y-axis limits make the figure hard to read. I understand they are the same across the figures and so need to accommodate the high values in panel c. But this causes the other panels to be squashed. Could the authors use a non-linear scale? Or maybe just cut them off at 20 instead and not in the caption that task 5 in panel c goes up to about 40 (it is my understanding that anything over 10 corresponds to lots of social learning, so going over 20 is not very meaningful).

Review form: Reviewer 2

Is the manuscript scientifically sound in its present form?

Yes

Are the interpretations and conclusions justified by the results?

Yes

Is the language acceptable?

Yes

Do you have any ethical concerns with this paper?

No

Have you any concerns about statistical analyses in this paper?

No

Recommendation?

Accept with minor revision (please list in comments)

Comments to the Author(s)

Please find my review in the attached document (Appendix A).

Decision letter (RSOS-201215.R0)

Dear Mr Saliveros,

On behalf of the Editors, we are pleased to inform you that your Manuscript RSOS-201215 "Learning strategies and long-term memory in Asian short-clawed otters (*Aonyx cinereus*)" has been accepted for publication in Royal Society Open Science subject to minor revision in accordance with the referees' reports. Please find the referees' comments along with any feedback from the Editors below my signature.

We invite you to respond to the comments and revise your manuscript. Below the referees' and Editors' comments (where applicable) we provide additional requirements. Final acceptance of

your manuscript is dependent on these requirements being met. We provide guidance below to help you prepare your revision.

Please submit your revised manuscript and required files (see below) no later than 7 days from today's (ie 09-Sep-2020) date. Note: the ScholarOne system will 'lock' if submission of the revision is attempted 7 or more days after the deadline. If you do not think you will be able to meet this deadline please contact the editorial office immediately.

on behalf of Dr Claudia Wascher (Associate Editor) and Kevin Padian (Subject Editor)
openscience@royalsociety.org

Associate Editor Comments to Author (Dr Claudia Wascher):

Thank you for submitting this paper for consideration to Royal Society Open Science. We have received comments from two reviewers, and they find the study interesting and well presented. The reviewers provide a number of valuable comments, intended to further improve clarity of the manuscript, which we recommend to carefully incorporate prior to publication. We are looking forward to reading the revised version of the study.

Reviewer comments to Author:

Reviewer: 1
Comments to the Author(s)

The authors describe a two-phase experiment studying the social learning of extractive foraging techniques in three groups of captive Asian short-clawed otters. They find good evidence that a tendency to interact with the task spreads socially, but weaker evidence that solving the task spreads in this way, suggesting solving the task relies on asocial learning to be solved. In addition, the otters solved the tasks faster a few months later, suggesting they remembered how to solve them. Older otters were more skilled at solving the tasks, as were males, and the otters probably found the harder tasks harder to solve.

Overall, I think this paper is strong. As the above summary suggests, the authors ask a lot of questions, they nonetheless provide compelling answers through a good experimental design and powerful analytic strategy. I have only minor concerns with the manuscript which are as follows:

lines 28-31 (and throughout). Here the authors set up social and asocial learning as two separate mechanisms. I know this isn't a focus of the paper, but I wonder if the emphasis on the

separateness is excessive. Yes, talking about social vs. asocial learning has been wonderfully productive in the scientific literature. But are they this separate at the psychological level, or is this just how we choose to think about it? We could equally ask “how sensitive are animals to social cues?” which is more nuanced (and fits perfectly with the analytical approach). So, no big deal, but maybe tweak the wording a bit?

lines 45-46. This sentence feels like it should be at the start of the next paragraph, not the end of this one.

lines 50-58. This paragraph needs better examples. Those provided are good examples of animals engaging in strategic social learning, but do we have proof this improves their efficiency and success?

lines 93-95. It feels odd that the notion of otters observing each other’s behavior isn’t raised as a possibility here.

lines 126-128. This is a good method. I’d additionally like a brief note that (1) there is no autocorrelation over the series of observations for each dyad, just so we know the observations are statistically independent (might be tricky to calculate this with only 15 observations though) and (2) there is significant variation between the association strengths of different dyads (i.e. do otters have structured social networks?).

lines 133. Do we really need this equation? It’s just describing a proportion. I think it would be easier to stick with words here.

lines 142-143. Are the authors not worried that removing one otter seemed to change the social network so drastically? I’d be worried that the social network is just noise. The authors results suggest social networks are real (they guide information flow), but still, this should probably be discussed in the discussion.

lines 145-147. Given that the social networks are apparently very labile, are the authors not worried that the reintroduction of the recovered otters produced yet another, novel, social network structure?

line 197 – typo: analyses

lines 216-224: This nomenclature (additive vs. multiplicative) feel weird. Why does additive mean that individual effects act only on asocial learning? I can accept that this is the case because of how NBDA works under-the-hood, but it’s not clear to me why this is the case.

lines 228-239: This paragraph is really hard to follow. It’s clear the authors did a lot of stuff, but I have little idea exactly what, or how this paragraph corresponds to the structure of the results section. What are the types of social learning models? What are the information types? etc.

Figure 3. The y-axis limits make the figure hard to read. I understand they are the same across the figures and so need to accommodate the high values in panel c. But this causes the other panels to be squashed. Could the authors use a non-linear scale? Or maybe just cut them off at 20 instead and not in the caption that task 5 in panel c goes up to about 40 (it is my understanding that anything over 10 corresponds to lots of social learning, so going over 20 is not very meaningful).

Reviewer: 2

Comments to the Author(s)

Please find my review in the attached document.

===PREPARING YOUR MANUSCRIPT===

- one version identifying all the changes that have been made (for instance, in coloured highlight, in bold text, or tracked changes);
- a 'clean' version of the new manuscript that incorporates the changes made, but does not highlight them.

 This version will be used for typesetting.

===PREPARING YOUR REVISION IN SCHOLARONE===

- Any electronic supplementary material (ESM).
- If you are requesting a discretionary waiver for the article processing charge, the waiver form must be included at this step.
- If you are providing image files for potential cover images, please upload these at this step, and inform the editorial office you have done so. You must hold the copyright to any image provided.
- A copy of your point-by-point response to referees and Editors. This will expedite the preparation of your proof.

- Ensure that your data access statement meets the requirements at <https://royalsociety.org/journals/authors/author-guidelines/#data>. You should ensure that you cite the dataset in your reference list. If you have deposited data etc in the Dryad repository, please only include the 'For publication' link at this stage. You should remove the 'For review' link.
- If you are requesting an article processing charge waiver, you must select the relevant waiver option (if requesting a discretionary waiver, the form should have been uploaded at Step 3 'File upload' above).
- If you have uploaded ESM files, please ensure you follow the guidance at <https://royalsociety.org/journals/authors/author-guidelines/#supplementary-material> to include a suitable title and informative caption. An example of appropriate titling and captioning may be found at https://figshare.com/articles/Table_S2_from_Is_there_a_trade-off_between_peak_performance_and_performance_breadth_across_temperatures_for_aerobic_scope_in_teleost_fishes_/3843624.

Author's Response to Decision Letter for (RSOS-201215.R0)

See Appendix B.

Decision letter (RSOS-201215.R1)

Dear Mr Saliveros,

It is a pleasure to accept your manuscript entitled "Learning strategies and long-term memory in Asian short-clawed otters (*Aonyx cinereus*)" in its current form for publication in Royal Society Open Science.

You can expect to receive a proof of your article in the near future. Please contact the editorial office (openscience_proofs@royalsociety.org) and the production office (openscience@royalsociety.org) to let us know if you are likely to be away from e-mail contact -- if

you are going to be away, please nominate a co-author (if available) to manage the proofing process, and ensure they are copied into your email to the journal.

on behalf of Dr Claudia Wascher (Associate Editor) and Kevin Padian (Subject Editor)
openscience@royalsociety.org

Appendix A

Review 'Learning strategies and long-term memory in Asian short-clawed otters (Aonyx cinereus)

This paper investigates whether Asian short-clawed otters rely on social learning to approach and solve novel foraging tasks, and whether they are capable of storing the acquired skills in their long-term memory. The authors exposed three captive groups of otters to 5 foraging tasks (of different difficulty levels) and used network-based diffusion analysis to test whether the otters relied on social information to i) approach and ii) solve the tasks. They then repeated the experiments several months later to see whether individuals remembered how to solve the tasks. They found that otters relied on social information (measured by testing whether the spread followed the association networks) to approach tasks, but less so to find a solution to the tasks, indicating the most likely mechanism was stimulus enhancement. Furthermore, they found that otters solved the tasks faster in the second presentation round, indicating that they are capable of long-term memory of learned skills.

In my opinion, this represents an interesting and thoroughly written study. The experimental and statistical procedures are well explained. The study is based on Ladds et al (2017)*, but refines methodology and analytical approaches, and brings in a new element of long-term memory of the acquired tasks. Overall, I only have a few minor comments, but suggest some clarifications on the interpretation of results in the discussion. Detailed comments follow here:

*Ladds, Z., Hoppitt, W., & Boogert, N. J. (2017). Social learning in otters. *Royal Society Open Science*, 4(8). <https://doi.org/10.1098/rsos.170489>

Abstract:

L 12-14 I suggest phrasing this with more care: Committing acquired information to long-term memory is only beneficial as long the information does not become outdated, but is not adaptive in changing environments or when dealing with ephemeral resources. I suggest changing to (or similar):

Furthermore, once animals have acquired new information, it can be beneficial for them to commit it to long-term memory, if it allows access to profitable resources in the future.

L20: suggest adding 'a capacity for social learning and long-term memory'

Introduction

L28 replace 'adjusted' with 'adapted'

L37 I think it is safe to omit '*Evidence is accumulating that*', as it has been shown repeatedly in recent years that animals use social learning strategies.

L59-60: I think it is important to state that this is not always the best strategy (same as in the abstract). Committing information to long-term memory is only useful when such information benefits future resource acquisition. I suggest adding this information.

L71: Suggest replacing '*most famed for*' with '*known/renowned for*'

L101: Suggest replacing '*honed*' with '*refined*'

Methods + Results

Nicely structured, easy to follow the relatively complex analysis.

L173 ...otters'...

Discussion

L339: Suggest replacing '*tackling*' with '*encountering*'

L343 ... *for the first time*... is redundant as you use 'novel' foraging task

L358-363: this paragraph might need some clarification. According to the result section (3.1.2 & partly in 3.2.2., although less clear of a result) the top models show that otters do rely on social information when finding the solutions to tasks if I understand correctly? Lines 359 and 360 contradict this directly ('... *may have relied more on asocial learning than on social learning to solve tasks.*'). If this was the case, wouldn't you expect an asocial model to outperform the social models? Can you clarify in how far you arrive at this conclusion based on the results of your models?

In this context, it may also be worth noting somewhere that there may some uncertainty about the strength of transmission of the spread of the task solutions. Looking at Table 3, the strength of social transmission for the solves is generally lower than for the interactions, but the large confidence intervals also indicate that these estimates have to be interpreted with caution. As far as I understand, the interpretation should be that otters seem to rely on social learning when finding solutions to the tasks, but the extent to which they do so is not quite clear.

One result was not discussed at all – the fact that the reliance on social learning varied between tasks but did not decrease or increase with task difficulty. Is it due to the fact that the perceived level of difficulty (from a human perspective) did not match the difficulty for the otters? Could the solve latencies be an indication for that, given that the most difficult task (task 5) ranked third? What implications does this result have for the learning strategies employed by the otters? And how do these results relate to findings in Ladds et al (2017). This would be interesting to add.

Tables, Figures, Data and SI

Table 2 and table 3: Happy to be corrected, but aren't these two tables essentially the same? Table 2 seems to contain the 'raw' estimates for the strength of social learning, while in table 3, these estimates

are converted to percentages (which in my opinion are easier to interpret). Unless there is good reason to retain table 2, I would consider removing it. Please correct me if I'm wrong here.

Data and R codes appear complete and with sufficient explanation.

Appendix B

Alex Saliveros, BSc

PhD student

Centre for Ecology and Conservation,
University of Exeter, Penryn Campus,
Penryn, Cornwall,
TR10 9FE

E-mail: ams243@exeter.ac.uk

Friday 2nd October 2020

Dear Editors of *Royal Society Open Science*,

Thank you very much for accepting our manuscript entitled '*Learning strategies and long-term memory in Asian short-clawed otters (Aonyx cinereus)*' for publication in *Royal Society Open Science* subject to minor revisions. It is very welcome news in this difficult and unusual time!

Please find an outline of the changes made below, in addition to a new version of the manuscript where revisions have been made using tracked changes.

Yours sincerely on behalf of all authors,

Alex Saliveros

Associate Editor Comments to Author (Dr Claudia Wascher):

Thank you for submitting this paper for consideration to Royal Society Open Science. We have received comments from two reviewers, and they find the study interesting and well presented. The reviewers provide a number of valuable comments, intended to further improve clarity of the manuscript, which we recommend to carefully incorporate prior to publication. We are looking forward to reading the revised version of the study.

Dear Dr. Claudia Wascher, thank you so much for your positive assessment of our manuscript! We hope you will find the revised version suitable for publication.

Comments from reviewer 1:

The authors describe a two-phase experiment studying the social learning of extractive foraging techniques in three groups of captive Asian short-clawed otters. They find good evidence that a tendency to interact with the task spreads socially, but weaker evidence that solving the task spreads in this way, suggesting solving the task relies on asocial learning to be solved. In addition, the otters solved the tasks faster a few months later, suggesting they remembered how to solve them. Older otters were more skilled at solving the tasks, as were males, and the otters probably found the harder tasks harder to solve.

Overall, I think this paper is strong. As the above summary suggests, the authors ask a lot of questions, they nonetheless provide compelling answers through a good experimental design and powerful analytic strategy. I have only minor concerns with the manuscript which are as follows:

We thank the reviewer for their positive and constructive feedback on our manuscript!

lines 28-31 (and throughout). Here the authors set up social and asocial learning as two separate mechanisms. I know this isn't a focus of the paper, but I wonder if the emphasis on the separateness is excessive. Yes, talking about social vs. asocial learning has been wonderfully productive in the scientific literature. But are they this separate at the psychological level, or is this just how we choose to think about it? We could equally ask "how sensitive are animals to social cues?" which is more nuanced (and fits perfectly with the analytical approach). So, no big deal, but maybe tweak the wording a bit?

We agree with the reviewer and have reworded the first paragraph (lines 29-32) to introduce social information use as being on a spectrum, rather than asocial and social learning as two separate mechanisms.

lines 45-46. This sentence feels like it should be at the start of the next paragraph, not the end of this one.

We have moved this sentence to the start of the next paragraph as suggested (line 51).

lines 50-58. This paragraph needs better examples. Those provided are good examples of animals engaging in strategic social learning, but do we have proof this improves their efficiency and success?

We have removed the previous examples and reworked this paragraph to include new examples that provide more definitive evidence that social learning occurs when asocial learning is costly and improves foraging efficiency and success (lines 56-61).

lines 93-95. It feels odd that the notion of otters observing each other's behavior isn't raised as a possibility here.

We have added "it may be that they attract each other to profitable food patches, and observe the food extraction techniques of their group mates" (line 107).

lines 126-128. This is a good method. I'd additionally like a brief note that (1) there is no autocorrelation over the series of observations for each dyad, just so we know the observations are statistically independent (might be tricky to calculate this with only 15 observations though) and (2) there is significant variation between the association strengths of different dyads (i.e. do otters have structured social networks?).

(1) The observations going into constructing the network do not need to be statistically independent for the purposes of NBDA, nor does a lack of statistical independence make the estimates of association themselves invalid; the number of sampling periods does not enter into the Network-Based Diffusion Analysis (NBDA), so there is no way in which use of non-independent points to create the social network would result in a pseudo-replication effect due to somehow artificially inflating sample size (lines 153-156). In addition, while the previous paper on social learning in Asian short-clawed otters by Ladds et al (2017 RSOS) collected 15 hours of social association data over 5 consecutive days, we doubled this observation period to 10 days. We would argue that this extended observation period should provide an accurate representation of the otter groups' social networks. This is confirmed by our results showing that the social networks predict information spread through the otter groups.

(2) We have now included a social differentiation coefficient for each otter social network (lines 174-181). This is a coefficient of variation between all the association indices that make up a particular network. It is calculated by dividing the standard deviation of all the association indices by the mean of the association indices and provides an indication of heterogeneity of the associations within the network. The social differentiation coefficients for each of the networks (Newquay twelve-otter network: 0.48; Newquay eight-otter network: 0.41; Tamar: 0.36; New Forest four-otter network: 0.52; New Forest five-otter network: 0.58) indicate that the strength of associations varied between otter dyads, and thus that the networks had a heterogeneous structure. Social differentiation coefficients have also been added to the captions of Figures S1 and S2 in the supplementary material.

lines 133. Do we really need this equation? It's just describing a proportion. I think it would be easier to stick with words here.

We have removed the equation to calculate association indices and now describe it in words instead (lines 146-151).

lines 142-143. Are the authors not worried that removing one otter seemed to change the social network so drastically? I'd be worried that the social network is just noise. The authors results suggest social networks are real (they guide information flow), but still, this should probably be discussed in the discussion.

Although the removal and reintroduction of the otter may have modified the original pre-removal five-otter network, the fact that the NBDA using both networks provided significant evidence for social transmission (see Results) suggests that these networks were representative of the group's social structure. If this were not the case, we would not expect to find any correlation between the network and the order of information acquisition. More generally, the more 'noise' the networks contain, the more conservative estimates of social transmission become. Thus, the detection of patterns in the data consistent with social transmission following the network suggests that the networks are meaningful

in that they approximate the pathways of transmission. Given that the current Discussion is already rather lengthy, we have added a paragraph explaining the above to section 1 of the Supplementary Materials.

lines 145-147. Given that the social networks are apparently very labile, are the authors not worried that the reintroduction of the recovered otters produced yet another, novel, social network structure?

We hope that our explanation above, and added to the Supplementary Materials Section 1, addresses this concern.

line 197 – typo: analyses

This typo has been corrected (line 226), thank you for spotting it!

lines 216-224: This nomenclature (additive vs. multiplicative) feel weird. Why does additive mean that individual effects act only on asocial learning? I can accept that this is the case because of how NBDA works under-the-hood, but it's not clear to me why this is the case.

We have clarified this by adding that in additive models, the asocial and social learning rate estimates are summed (line 246), while in multiplicative models they are multiplied (line 250).

lines 228-239: This paragraph is really hard to follow. It's clear the authors did a lot of stuff, but I have little idea exactly what, or how this paragraph corresponds to the structure of the results section. What are the types of social learning models? What are the information types? etc.

We have added a sentence re-iterating what the social learning model types are (lines 258-259), to further clarify the analyses. In addition, rather than referring to interacting and solving tasks as information types, which may have caused confusion, we now refer to them as behaviours (line 260, and throughout). We hope that these additions, and the references to the papers and manuals describing the NBDA method in detail, help to clarify this analytical process.

Figure 3. The y-axis limits make the figure hard to read. I understand they are the same across the figures and so need to accommodate the high values in panel c. But this causes the other panels to be squashed. Could the authors use a non-linear scale? Or maybe just cut them off at 20 instead and not in the caption that task 5 in panel c goes up to about 40 (it is my understanding that anything over 10 corresponds to lots of social learning, so going over 20 is not very meaningful).

We thank the reviewer for pointing this out. Y axis limits have now been cut off at 20 across all panels in order to improve legibility for those panels where social transmission rate estimates and respective confidence intervals are relatively low (line 621), namely panels (b) and (d). A note has been made in the caption that the upper 95% confidence interval of the social transmission rate estimate for when otters were interacting with task type 5 in the second task presentation round is 31.00, as the axis no longer extends high enough to accommodate the respective error bar (lines 629-631).

Note: While editing the figure we noticed that the error bars corresponded to confidence intervals from a previous iteration of our analyses. This mistake has been corrected and the error bars now correspond to the 95% confidence intervals of the social transmission rate estimates reported in the results section and Table 2. All other figures and tables were also checked but no other mistakes of this nature, or any others, were found.

Comments from reviewer 2:

Review 'Learning strategies and long-term memory in Asian short-clawed otter (*Aonyx cinereus*)

This paper investigates whether Asian short-clawed otters rely on social learning to approach and solve novel foraging tasks, and whether they are capable of storing the acquired skills in their long-term memory. The authors exposed three captive groups of otters to 5 foraging tasks (of different difficulty levels) and used network-based diffusion analysis to test whether the otters relied on social information to i) approach and ii) solve the tasks. They then repeated the experiments several months later to see whether individuals remembered how to solve the tasks. They found that otters relied on social information (measured by testing whether the spread followed the association networks) to approach tasks, but less so to find a solution to the tasks, indicating the most likely mechanism was stimulus enhancement. Furthermore, they found that otters solved the tasks faster in the second presentation round, indicating that they are capable of long-term memory of learned skills.

In my opinion, this represents an interesting and thoroughly written study. The experimental and statistical procedures are well explained. The study is based on Ladds et al (2017)*, but refines methodology and analytical approaches, and brings in a new element of long-term memory of the acquired tasks.

We thank the reviewer for their positive feedback and constructive suggestions for further clarification.

Overall, I only have a few minor comments, but suggest some clarifications on the interpretation of results in the discussion. Detailed comments follow here:

*Ladds, Z., Hoppitt, W., & Boogert, N. J. (2017). Social learning in otters. Royal Society Open Science, 4(8). <https://doi.org/10.1098/rsos.170489>

Abstract:

L 12-14 I suggest phrasing this with more care: Committing acquired information to long-term memory is only beneficial as long the information does not become outdated, but is not adaptive in changing environments or when dealing with ephemeral resources. I suggest changing to (or similar):

Furthermore, once animals have acquired new information, it can be beneficial for them to commit it to long-term memory, if it allows access to profitable resources in the future.

Thank you for this suggestion, we have added this phrasing to the abstract (lines 14-15).

L20: suggest adding 'a capacity for social learning and long-term memory'

We have changed this sentence accordingly (line 21. The suggested edits extended the abstract to surpass the word limit, so we have removed words in other places. We hope these edits have not reduced the clarity of the abstract.

Introduction

L28 replace 'adjusted' with 'adapted'

We respectfully decline to make this edit, as 'adapted' signifies genetic changes due to natural selection across evolutionary time to many biologists. However, we can make this change if requested by the editor.

L37 I think it is safe to omit 'Evidence is accumulating that', as it has been shown repeatedly in recent years that animals use social learning strategies.

We have removed the phrase 'Evidence is accumulating that' from the opening of the paragraph (line 42).

L59-60: I think it is important to state that this is not always the best strategy (same as in the abstract). Committing information to long-term memory is only useful when such information benefits future resource acquisition. I suggest adding this information.

We have reworked the sentence to denote the fact that committing information regarding foraging to long-term memory is only beneficial if it benefits future resource acquisition (lines 70-71).

L71: Suggest replacing 'most famed for' with 'known/reowned for'

We have made the suggested change to the terminology (line 83).

L101: Suggest replacing 'honed' with 'refined'

We have made the suggested change to the terminology (line 114).

Methods + Results

Nicely structured, easy to follow the relatively complex analysis.

Thank you!

L173 ...otters' ...

We have corrected the grammatical error (line 202), thank you for spotting it!

Discussion

L339: Suggest replacing 'tackling' with 'encountering'

We have made the suggested change to terminology (line 376).

L343 ... *for the first time*... is redundant as you use 'novel' foraging task

We have removed the redundant phrasing (line 380).

L358-363: this paragraph might need some clarification. According to the result section (3.1.2 & partly in 3.2.2., although less clear of a result) the top models show that otters do rely on social information when finding the solutions to tasks if I understand correctly? Lines 359 and 360 contradict this directly ('... may have relied more on asocial learning than on social learning to solve tasks.'). If this was the case, wouldn't you expect an asocial model to outperform the social models? Can you clarify in how far you arrive at this conclusion based on the results of your models?

We have included four broad types of NBDA models in our analysis of the puzzle box experiments: Additive social models, multiplicative models, no ILV models, and additive asocial models. Additive models assume ILVs only affect asocial learning rates so asocial and social learning are independent processes, and no ILV models assume ILVs affect neither asocial and social learning rates. Multiplicative models assume ILVs affect both social and asocial learning rates, so when a multiplicative model is the best overall model (like we have here), NBDA provides most support for the presence of a combination of both learning strategies (see lines 245-257 in methods section 2.6.1). Social learning can be present at a lower level than asocial learning, and vice versa. In this case, social learning is present but at a relatively low level (as indicated by the estimated social transmission rates from the best model ranging from 0.27 to 4.21 as compared to social transmission rates estimated to be 5.51 for the spread of task interactions). We therefore infer that asocial learning may have been

relatively important when otters are solving tasks in the first round of task presentations. An asocial model would only be best supported if there was no model support for social learning at all.

We have tried to make this clear in the methods and results section:

Lines 248-252: “In contrast, multiplicative models are fitted with parameters that assume ILVs affect both asocial and social learning rates, so when these models best fit the data, new behaviours are inferred to be acquired via a combination of both learning methods”.

Lines 316-317: “This suggests that, like with their first task interactions (above), otters’ first task solves were attributable to a mix of both social and asocial learning”.

Lines 346-347: “This suggests that the first task solves achieved by the otters in the second round of task presentations were attributable to a mix of both social and asocial learning”.

We have also rephrased the sentence in the discussion (mentioned above) to further clarify this (lines 396-398). Instead of stating “may have relied more on asocial learning than on social learning to solve tasks”, we have changed this to “they are less reliant on social information when actually learning to solve tasks”. This highlights that, although social learning plays less of a role when otters are learning to solve tasks, it is still present.

In this context, it may also be worth noting somewhere that there may be some uncertainty about the strength of transmission of the spread of the task solutions. Looking at Table 3, the strength of social transmission for the solves is generally lower than for the interactions, but the large confidence intervals also indicate that these estimates have to be interpreted with caution. As far as I understand, the interpretation should be that otters seem to rely on social learning when finding solutions to the tasks, but the extent to which they do so is not quite clear.

We have removed the sentence “Where otters relied on social learning, they solved tasks by copying individuals closely associated to them” from results section “3.1.2 Task solving - Round 1” in order to soften the language around the findings of otters using social learning to solve tasks in the first presentation round (lines 321-322).

Lines 317-321 and Lines 356-359: We have now reported the wide range of the 95% confidence intervals for the percentage of solves due to social learning in round one. Thus, although the best model suggests social learning is present, the extent to which otters use it to solve tasks is unclear.

One result was not discussed at all – the fact that the reliance on social learning varied between tasks but did not decrease or increase with task difficulty. Is it due to the fact that the perceived level of difficulty (from a human perspective) did not match the difficulty for the otters? Could the solve latencies be an indication for that, given that the most difficult task (task 5) ranked third? What implications does this result have for the learning strategies employed by the otters? And how do these results relate to findings in Ladds et al (2017). This would be interesting to add.

We have added a section to the discussion to address these findings (lines 427-456). Given that otters’ social information use was highest with the easiest and hardest to solve tasks, we suggest they do no ‘copy when asocial learning is costly’. Regarding otters interacting with tasks, we suggest in the first round social information use is the same as they have no prior knowledge of task difficulty, but in the second round it increases with assumed task difficulty as otters remembered the difficulty of particular tasks and adjusted their reliance on social learning accordingly. However this finding should be interpreted cautiously as assumed task difficulty did not reflect true task difficulty according to the solve latencies.

Tables, Figures, Data and SI

Table 2 and table 3: Happy to be corrected, but aren't these two tables essentially the same? Table 2 seems to contain the 'raw' estimates for the strength of social learning, while in table 3, these estimates are converted to percentages (which in my opinion are easier to interpret). Unless there is good reason to retain table 2, I would consider removing it. Please correct me if I'm wrong here.

Table 3 contains percentage conversions of the raw social transmission rate estimates reported in Table 2, however we feel it is important to include both tables. Raw social transmission rate estimates are the more commonly reported metric of social learning in studies using NBDA. Therefore reporting such estimates here allows for comparisons between this study and others, as well as allowing for the potential inclusion of this study in future meta-analyses. Indeed, Table 3 was only included to consolidate the results reported in Table 2 as a way to aid interpretation for more casual readers. Furthermore, Table 2 is valuable in supporting Figure 3 by providing the exact values displayed in the figure.

Data and R codes appear complete and with sufficient explanation.

Thanks!